# SARS-CoV-2 specific antibody and neutralization assays reveal the wide range of the humoral immune response to virus

Mikail Dogan[1], Lina Kozhaya[1], Lindsey Placek[1], Courtney Gunter[1], Mesut Yigit[1], Rachel Hardy[1], Matthew Plassmeyer[2], Paige Coatney[2], Kimberleigh Lillard[2], Zaheer Bukhari[3], Michael Kleinberg[4], Chelsea Hayes[5], Moshe Arditi[6], Ellen Klapper[5], Noah Merin[7], Bruce Tsan-Tang Liang[4], Raavi Gupta[3], Oral Alpan[2] & Derya Unutmaz[1,8 ✉]

Development of antibody protection during SARS-CoV-2 infection is a pressing question for public health and for vaccine development. We developed highly sensitive SARS-CoV-2-specific antibody and neutralization assays. SARS-CoV-2 Spike protein or Nucleocapsid protein specific IgG antibodies at titers more than 1:100,000 were detectable in all PCR+ subjects ($n = 115$) and were absent in the negative controls. Other isotype antibodies (IgA, IgG1-4) were also detected. SARS-CoV-2 neutralization was determined in COVID-19 and convalescent plasma at up to 10,000-fold dilution, using Spike protein pseudotyped lenti-viruses, which were also blocked by neutralizing antibodies (NAbs). Hospitalized patients had up to 3000-fold higher antibody and neutralization titers compared to outpatients or convalescent plasma donors. Interestingly, some COVID-19 patients also possessed NAbs against SARS-CoV Spike protein pseudovirus. Together these results demonstrate the high specificity and sensitivity of our assays, which may impact understanding the quality or duration of the antibody response during COVID-19 and in determining the effectiveness of potential vaccines.

[1] Jackson Laboratory for Genomic Medicine, Farmington, CT, USA. [2] Amerimmune, Fairfax, VA, USA. [3] SUNY Downstate Medical Center, Department of Pathology, Brooklyn, NY, USA. [4] Calhoun Cardiology Center, University of Connecticut School of Medicine, Farmington, CT, USA. [5] Department of Pathology & Laboratory Medicine and Transfusion Medicine, Cedars-Sinai Medical Center, Los Angeles, CA, USA. [6] Department of Pediatric, Division of Pediatric Infectious Diseases and Immunology, Biomedical Sciences, Cedars-Sinai Medical Center, Los Angeles, CA, USA. [7] Department of Internal Medicine, Division of Hematology Cedars-Sinai Medical Center, Los Angeles, CA, USA. [8] Department of Immunology, University of Connecticut School of Medicine, Farmington, CT, USA. ✉email: derya@mac.com

Severe acute respiratory syndrome coronavirus 2 (SARS-CoV-2), which has caused the COVID-19 pandemic, enters target cells through the interaction of its envelope spike protein with the primary host cell receptor angiotensin-converting enzyme-2 (ACE2), which is then cleaved by a serine protease (TMPRSS2) to allow viral fusion and entry across the cell membrane[1]. Antibodies that can bind to the spike protein have the potential to neutralize viral entry into cells and are thought to play an important role in the protective immune response to SARS-CoV-2 infection[2–11].

To predict protection against SARS-CoV-2, it is critical to understand the quantity, quality and duration of the antibody response during different stages of COVID-19 and in the convalescent period. In this regard, assessing the level of neutralizing antibodies (NAbs) that block viral entry into cells could be a critical parameter in determining protection from SARS-CoV-2 and management of convalescent plasma therapies, which are being tested as a COVID-19 treatment option[12–15]. Defining the relationship between disease severity, other individual-specific comorbidities and the NAb response will be critical in our understanding of COVID-19 and in tailoring effective therapies.

Currently available SARS-CoV-2 antibody tests mostly lack sufficient dynamic range and sensitivity to allow for accurate detection or determination of the magnitude of the antibody response[16]. Furthermore, potential cross-reactivity among SARS-CoV-2 specific antibodies to other endemic coronaviruses could also be confounders in these tests[17–20], thus making them less reliable. Determining neutralization activity in patient plasma also has challenges, as these assays generally rely on live virus replication, requiring a high-level biohazard security BSL-3 level laboratory. Therefore, there is an unmet need to develop sensitive antibody and virus neutralization assays that are sufficiently robust for screening and monitoring large numbers of SARS-CoV-2 infected or convalescent subjects.

To overcome these experimental challenges, here we developed: (1) Highly sensitive bead-based fluorescent immunoassay for measuring SARS-CoV-2 specific antibody levels and isotypes, and (2) Robust SARS-CoV-2 spike protein pseudovirus to measure NAb levels in COVID-19 patient plasma. We found striking differences in total antibody levels and neutralization titers between hospitalized or severe COVID-19 patients relative to outpatient or convalescent plasma donors, which were obtained with the purpose of transfer to and treatment of patients. Significant correlations between antibody levels and neutralization titers, age and NAbs to SARS-CoV were also observed. These assays and findings have important implications for assessing the breadth and depth of the humoral immune response during SARS-CoV-2 infection and for the development of effective antibody-based therapies or vaccines.

## Results

### Development of SARS-CoV-2 specific antibody assay. Determining antibody response in SARS-CoV-2 infected subjects remains challenging, due to lack of sufficient dynamic range to determine precise antibody titers with antibody isotypes simultaneously. To overcome these obstacles, we developed a fluorescent bead-based immunoassay that takes advantage of the high dynamic range of fluorescent molecules using flow cytometry (Fig. 1a). In this assay, we immobilized biotinylated SARS-CoV-2 spike protein receptor-binding domain (RBD) or the Nucleoprotein (N) on streptavidin beads to detect specific antibodies from patient plasma (Fig. 1a). Different antibody isotypes were measured using anti-Ig (IgG, IgA, IgM, IgG1-4) specific secondary antibodies conjugated to a fluorescent tag (Fig. 1a). Using either anti-S-RBD antibody or soluble ACE2-Fc, we show very

high sensitivity in detecting spike protein binding, down to picogram ranges (Fig. 1b). Similarly, S-RBD-specific antibodies were detectable in serial dilutions up to 100,000-fold of plasma samples from SARS-CoV-2 PCR+ subjects at high specificity and sensitivity (Fig. 1c). We then used the titration curves from COVID-19 convalescent and healthy control plasma to normalize the area under the curve (AUC) values to quantitate the antibody levels (Supplementary Fig. 1a). Negative threshold values were set using healthy control AUC levels plus one standard deviation of the mean.

In addition to S-RBD and Nucleocapsid protein, we also attached different viral components such as S1 subunit of spike protein, S1 subunit N terminal domain (NTD) and S2 extracellular domain (ECD) onto the magnetic beads and tested IgG levels specific to those viral proteins to compare the antibody levels they detect. Interestingly, S-RBD captured significantly more antibodies compared to S1, which is the subunit of spike protein that contains S-RBD ($p = 0.0356$) (Fig. 1d).

We also evaluated the dynamic range of our assay by screening some of the plasma samples with a commercial ELISA-based antibody assay next to our bead-based assay and comparing the detected antibody levels. Antibody levels from the two antibody assays showed a high correlation ($r_s = 0.86$), confirming our assay's precision, and the bead-based antibody assay showed a wider dynamic range compared to the ELISA-based assay (Fig. 1e).

Using the bead-based assay, we screened COVID-19 patient or convalescent plasma samples (Table 1; $n = 115$) for total S-RBD and Nucleocapsid specific IgG AUC values of COVID-19 positive subjects, which varied 3-logs from $\sim 10^4$ to $\sim 10^7$ (Fig. 2a). S-RBD-specific IgM (40/40) and IgA (115/115) were also detectable and above the negative control threshold in all subjects (Fig. 2a). Statistical sensitivity and specificity estimates of our bead-based antibody assays were 100% and 99.34% for S-RBD IgG; 100% and 90.9% for S-RBD IgM; 94.26% and 87.87% for S-RBD IgA and 99.13% and 94.93% for Nucleocapsid IgG, respectively. Furthermore; S1 subunit, S1 N terminal domain (NTD), S2 extracellular domain (ECD) and Nucleocapsid protein-specific IgG and S-RBD-specific IgA levels positively correlated with S-RBD IgG antibodies (Supplementary Fig. 1b, c) with the highest correlation with S1 IgG ($r_s = 0.987$). Notably, IgG1 subclass antibody levels were comparable to total IgG levels whereas the other subtypes were relatively lower (Fig. 2b). There were significant differences in S-RBD or Nucleocapsid antibody levels between outpatient, hospitalized, and intensive care unit (ICU)/deceased subjects, with the highest levels observed in the most severe cases (Fig. 2c–e). Importantly, subjects who had recovered from COVID-19 and were also potential donors for convalescent plasma therapy (hereafter referred to as plasma donors), also had significantly lower antibody titers than hospitalized, ICU or deceased patients (Fig. 2c–e). Overall, individual S-RBD and Nucleocapsid IgG levels appeared to correlate with their IgA and IgG subclass (IgG1-4) responses to S-RBD (Supplementary Fig. 1d). Subdividing the subjects by sex did not reveal any statistical difference in IgG levels at any of the disease stages (Fig. 2f).

### Development of SARS-CoV-2 spike-protein pseudovirus. Next, we sought to develop a sensitive and high-throughput SARS-CoV-2 neutralization assay by incorporating SARS-CoV-2 spike protein into lentiviruses to assess specific inhibition of viral entry. To produce spike protein pseudotyped lentiviral particles, we first ensured expression of the spike protein on the cell membrane of transfected 293 cells, from which it would incorporate into the lentiviruses. Human codon-optimized SARS-CoV-2 spike protein

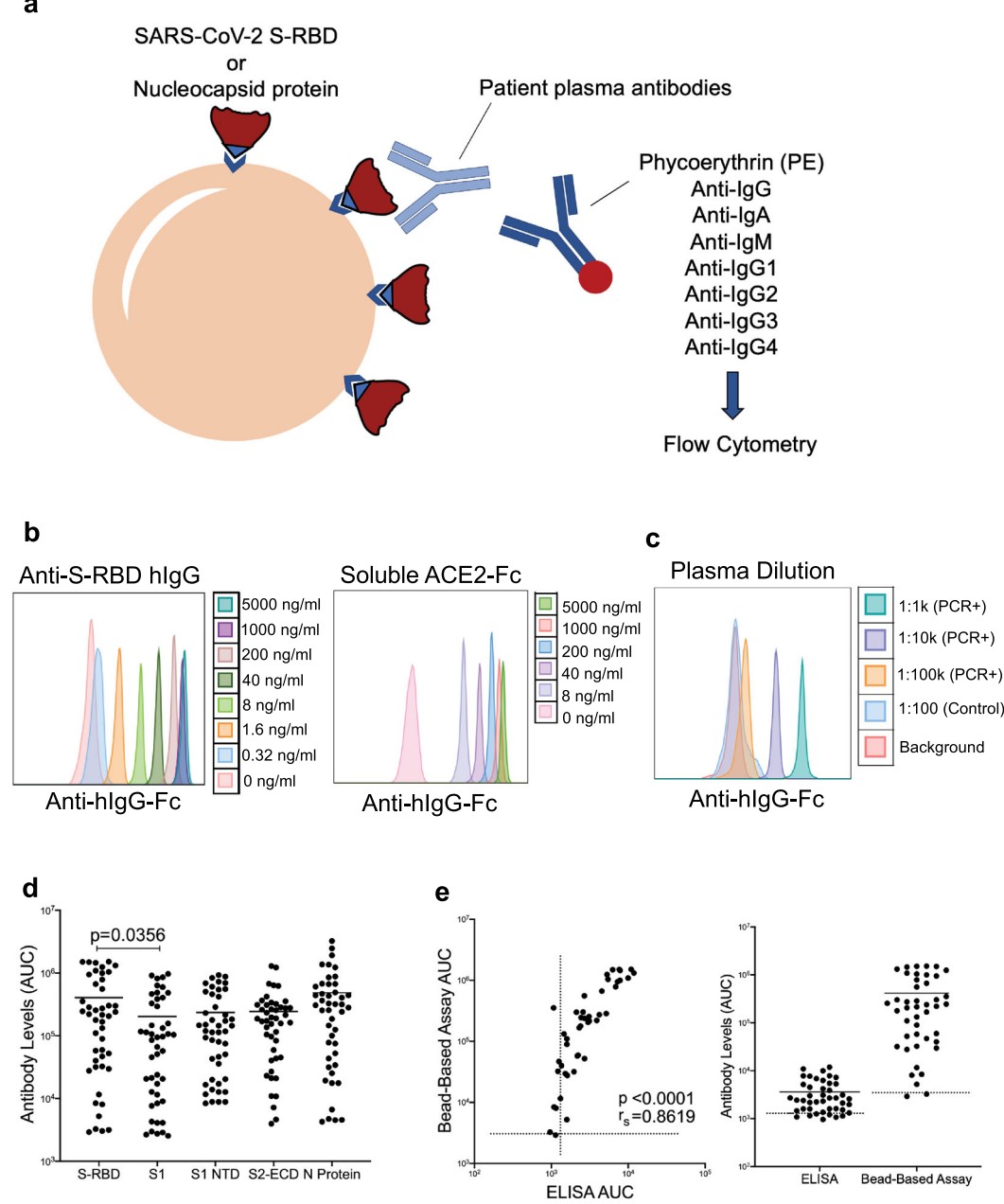

**Fig. 1 SARS-CoV-2 specific antibody detection assay. a** Illustration of antibody detection assay. Biotinylated S-RBD or Nucleocapsid proteins are captured by streptavidin-coated beads, then incubated with plasma samples and stained with PE-conjugated anti-IgG, IgA, IgM, IgG1, IgG2, IgG3, IgG4 antibodies. Fluorescence intensity analyzed by flow cytometry. **b** Histogram overlays demonstrating the detection of anti-S-RBD human IgG antibody (left) and soluble ACE2-Fc (right) as positive controls for plasma antibody assay. **c** Representative patient plasma titration. Healthy control plasma at 1:100 dilution was used as a negative control. Serial dilutions were used in the flow cytometry overlay. **d** Comparison of IgG antibody levels captured by S-RBD, S1 subunit of spike, S1 N terminal domain (NTD), S2 extracellular domain (ECD) and nucleocapsid protein coated beads ($n = 46$ biologically independent samples). **e** Correlation and comparison of bead-based assay S-RBD IgG antibody levels with ELISA-based assay ($n = 44$). Two-tailed Mann–Whitney $U$ test was used to determine the statistical significance in (**d**) and two-tailed Spearman's was used for correlation significance in (**e**). Horizontal bars in (**d**) and (**e**) indicate mean values.

sequences with and without endoplasmic reticulum retention signal (ERRS), which would be predicted to be more efficiently expressed on the cell surface membranes, were cloned into an expression vector and transfected into 293 cells. To evaluate membrane expression of spike protein, cells were stained with recombinant soluble ACE2-Fc fusion protein followed by a secondary staining with an anti-Fc antibody (Fig. 3a). The percentages of spike protein over-expressing cells were similar in the presence or absence of ERRS, but cells expressing spike protein without ERRS showed a higher geometric mean of expression (Fig. 3b). As such, we used spike protein lacking the ERRS for lentiviral pseudotyping to ensure its higher incorporation onto viral membranes.

We then co-transfected 293 cells with replication-defective lentivectors encoding GFP or RFP reporter genes and the spike protein-encoding plasmid and harvested the supernatant at 24 h, which was then used to infect cells expressing ACE2 (Fig. 3c). Bald particles were generated by transfecting lentivirus plasmids

**Table 1 Characteristics of SARS-CoV-2 infected and control subjects.**

| Demographics | | Healthy controls (n = 56) | Negative (n = 94) | Outpatient (n = 39) | Hospitalized (n = 19) | ICU/deceased (n = 24) | Plasma donors (n = 33) |
|---|---|---|---|---|---|---|---|
| Sex | Male | 14 | 33 | 11 | 7 | 10 | 18 |
| | Female | 42 | 61 | 28 | 12 | 14 | 15 |
| Age | Mean (±SEM) | 45.5 (±1.78) | 54.1 (±1.97) | 46.0 (±2.20) | 62.2 (±3.41) | 68.0 (±1.87) | 45.5 (±1.99) |
| | Median | 47.0 | 59.0 | 47.0 | 63.0 | 70.0 | 48.0 |
| Days between PCR/blood | Mean (±SEM) | N/A | N/A | 40.7 (±2.79) | 21.0 (±3.28) | 25.8 (±3.17) | 65.4 (±1.68) |
| | Median | N/A | N/A | 43.0 | 28.0 | 24.5 | 66.0 |

without any envelope and used as a negative control. Next, we tested the transduction efficiency of the viruses on wild-type 293 cells, given they express low levels of endogenous ACE2 (Supplementary Fig. 2a, b). While we found clearly defined infection of 293 cells with spike-protein pseudovirus compared to bald virions, infection rate determined by GFP or RFP expression was relatively low (Fig. 3d). We therefore generated human-ACE2 over-expressing 293 cells with a GFP reporter (ACE2-IRES-GFP) or fused to fluorescent mKO2 protein (ACE2-mKO2). ACE2 overexpression of ACE2-IRES-GFP or ACE2-mKO2 was confirmed by staining with SARS-CoV-2 spike-protein S1 subunit fused with mouse Fc (mFc) and anti-mFc secondary antibody (Supplementary Fig. 2a, b). Indeed, these ACE2 over-expressing 293 cells (293-ACE2) were efficiently transduced with spike protein pseudoviruses encoding either GFP or RFP (Fig. 3e). The efficiency of spike-protein pseudovirus infection was comparable in ACE2-IRES-GFP or ACE2-mKO2 fusion protein (Fig. 3e), and therefore both were used in subsequent neutralization experiments. In addition, we developed SARS-CoV spike protein pseudotyped lentivirus, which similarly infected 293-ACE2 cells at almost 100% efficiency at higher virus supernatant volumes (Fig. 3f). We also tested the stabilities of SARS-CoV-2 and SARS-CoV spike protein pseudotyped lentiviruses after serial freeze/ thaw cycles and found that their infectivity remained mostly similar with little loss of activity after 3 cycles (Fig. 3f).

**Neutralization of SARS-CoV-2 spike-protein pseudovirus with soluble ACE2, NAbs, and COVID-19 plasma**. We next investigated whether spike protein pseudoviruses could be neutralized by soluble ACE2 (sACE) or spike protein-specific NAbs (Fig. 4a). For this experiment, spike protein pseudotyped SARS-CoV-2 and SARS-CoV pseudoviruses were pre-cultured with different concentrations of sACE or NAbs, then added to 293-ACE2 cells. Subsequently, the infection levels were determined 3 days post-infection based on GFP or RFP expression as described above. sACE2 neutralized both SARS-CoV-2 and SARS-CoV pseudovirus infections in a dose-dependent manner (Fig. 4b, c), although neutralization of SARS-CoV-2 was slightly better than that of SARS-CoV pseudoviruses (Fig. 4b, c, and Supplementary Fig. 3a). Furthermore, spike-RBD-specific NAb neutralized SARS-CoV-2 pseudovirus entry much more efficiently than sACE2 but had no effect on SARS-CoV pseudovirus (Fig. 4c). One of the SARS-CoV-2 S-RBD-specific antibodies (non-NAb) did not show any neutralization of SARS-CoV-2, however very low-level neutralization of SARS-CoV pseudovirus was detected (Fig. 4c). We also observed measurable differences in the neutralizing activity of four different NAbs and two different soluble ACE2 proteins from different sources (Fig. 4d), showing the utility of this assay for such screening. Taken together, these experiments demonstrate that the combination of pseudotyped viruses and 293-ACE2 cells can be used to generate highly sensitive SARS-CoV-2 and SARS-CoV neutralization assays.

Using this approach, we then tested neutralization titers from COVID-19 patients or seropositive donors with a serial dilution of their plasma. Accordingly, plasma samples in threefold serial dilutions were incubated with spike pseudovirus and added to 293-ACE2 cells and infection was determined as described in Fig. 4. Healthy control plasma samples were used as negative controls whereas anti-S-RBD NAb served as a positive control (Fig. 5a). None of the control plasma (n = 34, 1 shown in Fig. 5a) tested showed any neutralization activity, whereas patient plasma efficiently neutralized the virus at up to 10,000-fold serial dilution (Fig. 5a). The 50% neutralization titer (NT50) was determined using the half-maximal inhibitory concentration values of plasma samples, normalized to control infections, from their serial

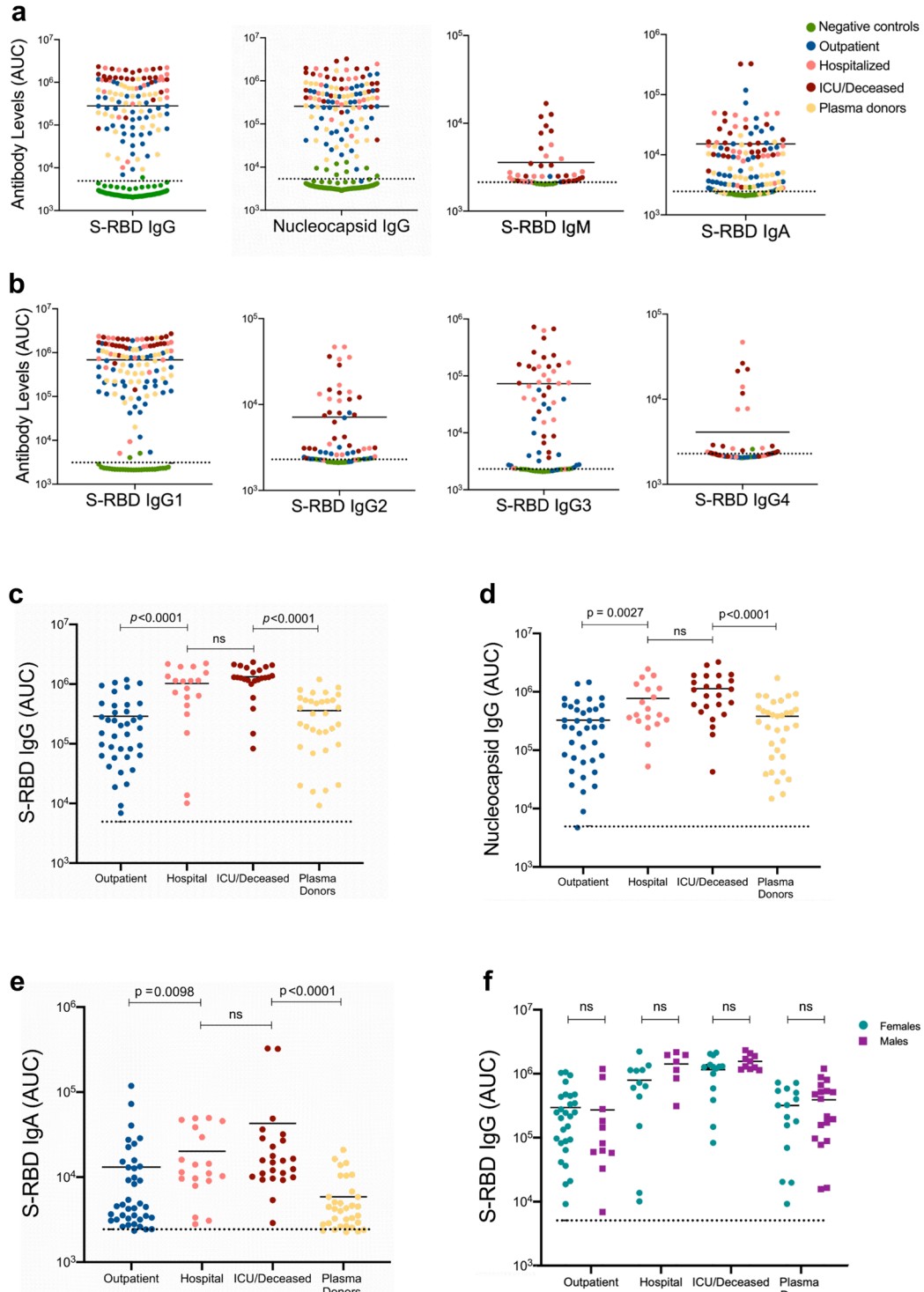

**Fig. 2 SARS-CoV-2 specific antibody detection in COVID-19 and convalescent plasma samples. a** Measurement of spike protein and nucleocapsid protein-specific IgG and spike protein-specific IgM and IgA antibodies as described in Fig. 1. Area under the curve (AUC) values of plasma antibodies were calculated from reciprocal dilution curves in antibody detection assay ($n = 256$ for S-RBD IgG and Nucleocapsid IgG, $n = 50$ for S-RBD IgM, $n = 144$ for S-RBD IgA). Dotted lines indicate the negative threshold calculated by adding 1 standard deviation to the mean AUC values of healthy controls' plasma. Horizontal bars show the mean value. Green, blue, salmon, red and yellow dots indicate negative controls, outpatient, hospitalized, ICU/deceased and plasma donor subjects, respectively. **b** S-RBD-specific IgG subclass AUC levels ($n = 144$ for S-RBD IgG1, $n = 74$ for S-RBD IgG2, S-RBD IgG3 and S-RBD IgG4) **c** S-RBD IgG AUC values of subject plasma grouped by outpatient, hospitalized, ICU or deceased and plasma donors ($n = 115$) **d** Nucleocapsid protein IgG AUC values of subject plasma grouped by outpatient, hospitalized, ICU or deceased and convalescent plasma donors ($n = 115$) **e** S-RBD IgA AUC values of subject plasma grouped by outpatient, hospitalized, ICU or deceased and plasma donors ($n = 115$). **f** S-RBD IgG AUC values of severity groups and plasma donors subdivided into males and females ($n = 115$). Green dots show female subjects while purple squares indicate male subjects. Statistical significances were determined using two-tailed Mann–Whitney U test.

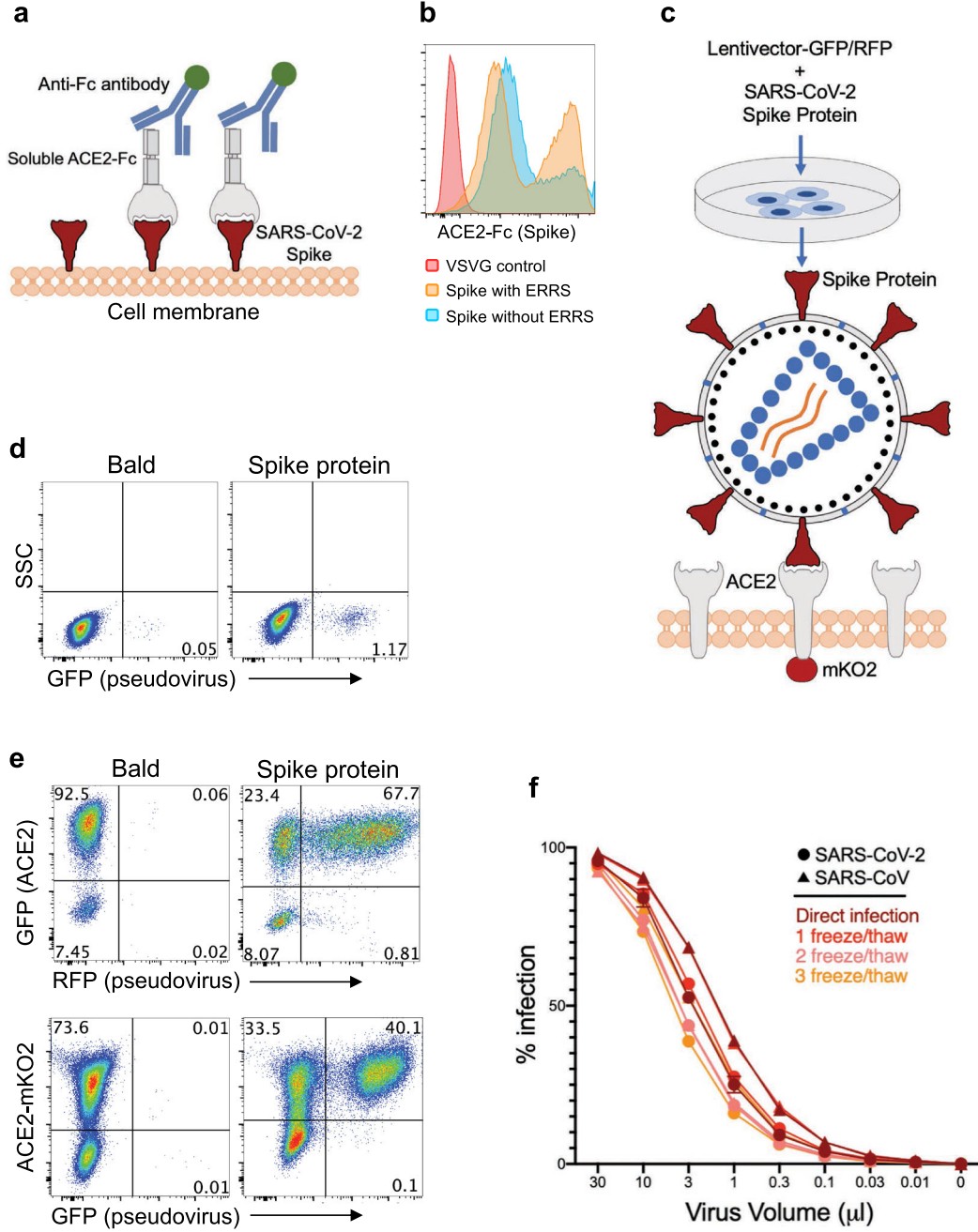

**Fig. 3 Development of SARS-CoV-2 and SARS-CoV spike-protein pseudotyped lentiviruses. a** Schematic illustration of spike protein expression on the cell surface and soluble ACE2-Fc staining followed by an anti-Fc antibody staining. **b** 293 cells transfected with spike protein with or without endoplasmic reticulum retention signal (ERRS) or with VSV-G as a negative control. The cells were stained with ACE2-Fc and anti-Fc-APC secondary antibody, flow cytometry data overlays are shown. **c** Schematic representation of spike protein pseudovirus generation and subsequent infection of ACE2-expressing host cells. A lentivector plasmid and a spike protein over-expressing envelope plasmid are used to co-transfect 293 cells to generate spike pseudovirus that in turn infect engineered cells over-expressing wild-type ACE2 or ACE2-mKO2 fusion. **d** Infection of wild-type 293 cells with either bald lentiviruses generated without envelope plasmid or spike protein pseudovirus. **e** Infection of 293-ACE2 cells with bald and spike lentiviruses. GFP and mKO2 markers are used to determine ACE2 over-expressing cells in ACE2-IRES-GFP and ACE2-mKO2, respectively. **f** The titrations of SARS-CoV-2 and SARS-CoV spike protein pseudoviruses encoding RFP. Triangles and circles represent SARS-CoV and SARS-CoV-2 data, respectively. Brown, red, salmon and orange-colored lines show direct infection, first, second and third freeze/thaw cycles, respectively. ACE2-IRES-GFP expressing 293 cells were incubated with threefold serial dilutions of virus supernatant, stored for several hours at 4 °C or serially frozen and thawed for 1, 2 and 3 cycles, and analyzed for RFP expression by flow cytometry on day 3 post-infection. Percent infection is % RFP+ cells after gating on GFP+ cells (i.e., ACE2+). Titration experiments were replicated twice except for the '1 freeze/thaw cycle' for which titrations were replicated four times. Error bars represent 1 standard deviation of mean values.

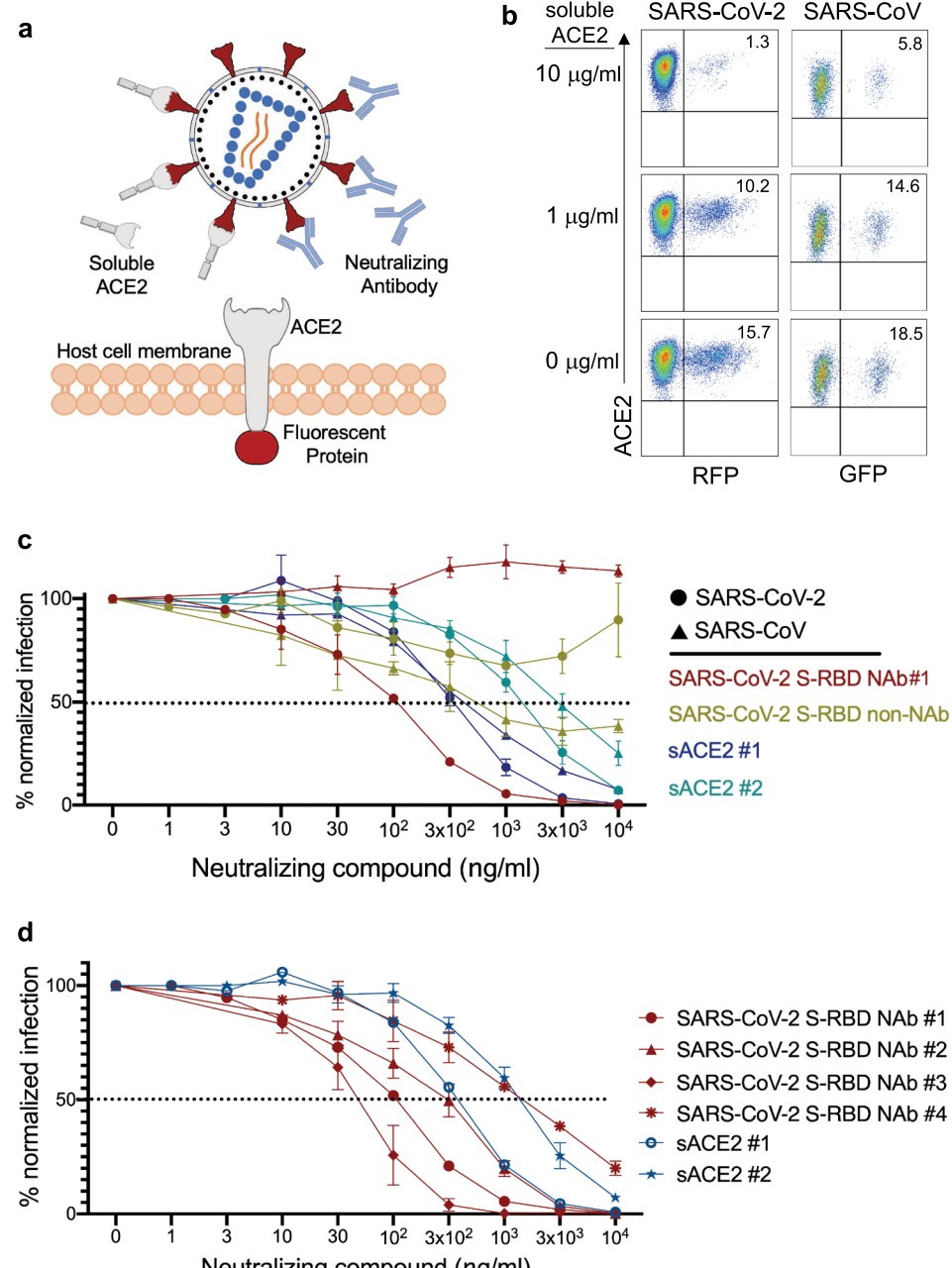

**Fig. 4 Neutralization of SARS-CoV-2 and SARS-CoV pseudoviruses with soluble ACE2 and Nabs. a** Illustration of spike-protein pseudovirus blocked by soluble ACE2 or neutralizing antibodies. **b** SARS-CoV-2 and SARS-CoV pseudovirus neutralization with soluble ACE2. SARS-CoV-2 RFP and SARS-CoV GFP pseudoviruses were pre-incubated with soluble ACE2 for 1 h and added to 293 cells expressing ACE2-IRES-GFP or ACE2-mKO2 fusion, respectively. **c** Neutralization of SARS-CoV-2 and SARS-CoV with S-RBD-specific antibodies and soluble ACE2 (sACE2). Viruses were pre-incubated with antibodies (NAb#1 and SARS-CoV-2 S-RBD non-NAb) or soluble ACE2 (sACE2) proteins for 1 h at the concentrations shown and subsequently added to target cells. Expression of RFP was determined at day 3 post-infection. Infection percentages were normalized to negative controls which are the infection conditions with no blocking agent. Triangles and circles represent SARS-CoV and SARS-CoV-2 data, respectively. Red, green, blue and turquoise colored lines show SARS-CoV-2 S-RBD NAb#1, SARS-CoV-2 S-RBD non-NAb, soluble ACE2 #1 and #2, respectively. **d** Neutralization of SARS-CoV-2 pseudoviruses using 4 different S-RBD NAbs and two different soluble ACE2 proteins. NAb #1 and #4 were human antibodies whereas NAb #2 and #3 were mouse. Red lines represent antibodies while blue lines show soluble ACE2 molecules. Dot, triangle, square, asterisk, circle and star symbols indicate SARS-CoV-2 S-RBD NAb #1, #2, #3, #4, soluble ACE2 #1 and #2, respectively. Graphs in (**c**) and (**d**) represent three replicates of the experiments. Error bars indicate one standard deviation of mean values.

dilutions. Importantly, the NT50 values of the subjects were much higher in hospitalized patients than in outpatients (Fig. 5b). NT50 values for hospitalized and ICU/deceased subjects were also up to 1000-fold higher than convalescent plasma donors (Fig. 5b). Hospitalized males and females, separately, also remained higher

in their NT50 levels and no difference was observed within each group (Fig. 5c).

We also tested whether SARS-CoV-2 PCR+ plasma could neutralize the SARS-CoV pseudovirus. In total, 104 plasma samples from all groups were tested for their ability to neutralize

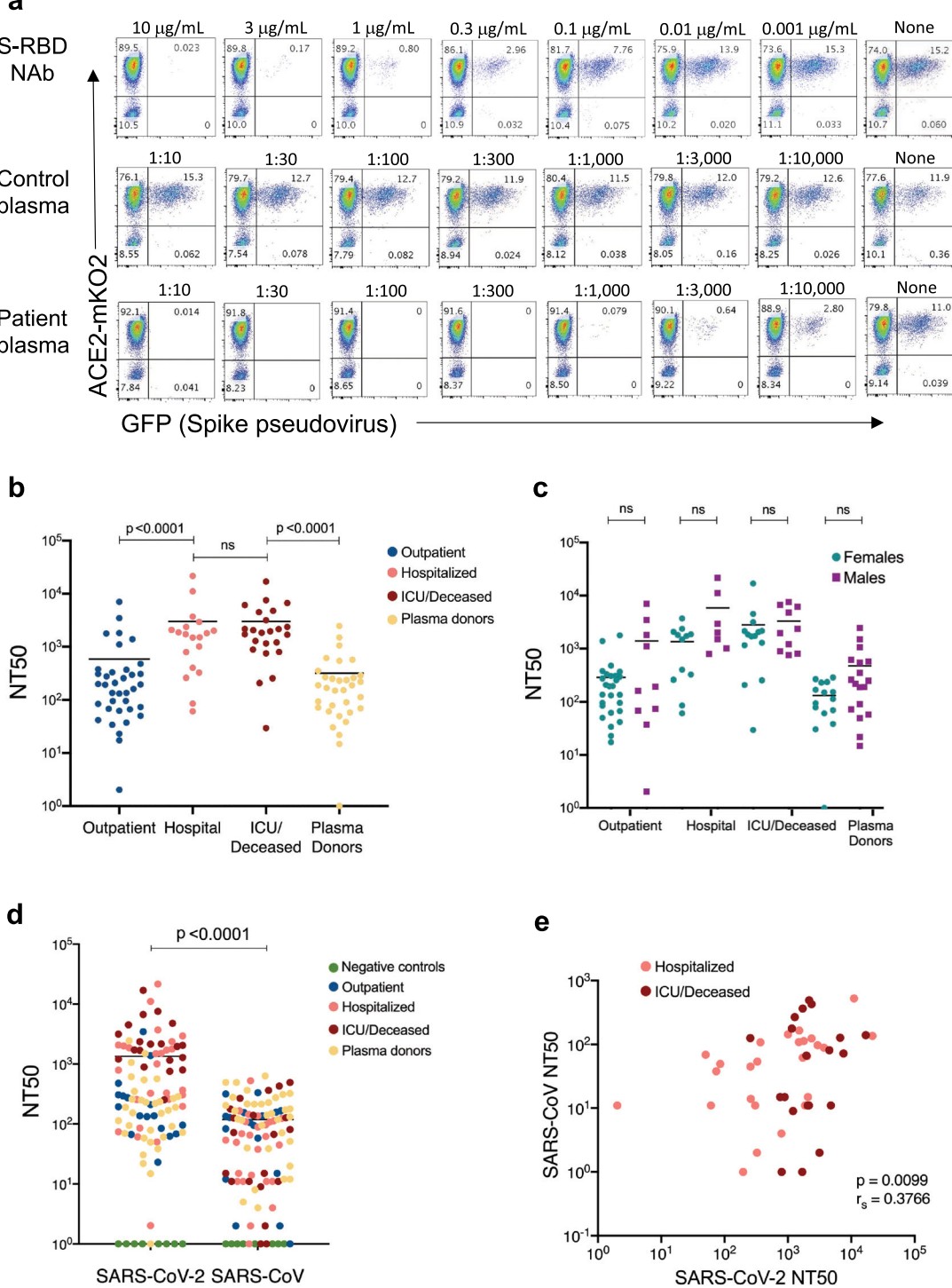

**Fig. 5 Neutralizing titers for SARS-CoV-2 and SARS-CoV in COVID-19 subject plasma. a** Neutralization assay with S-RBD-specific NAb, healthy control plasma, and a COVID-19 patient plasma. Threefold serial dilutions of NAb from 10 μg/ml to 1 ng/ml or the plasma from 1:10 to 1:10,000 were pre-incubated with spike protein pseudovirus and added to 293-ACE2 cells. GFP expression was analyzed by flow cytometry 3 days post infection. **b** SARS-CoV-2 neutralization titers (NT50) of COVID-19 plasma grouped as an outpatient, hospitalized, ICU or deceased and convalescent plasma donor groups (*n* = 113). **c** NT50 of COVID-19 patient and plasma donor groups subdivided into males and females (*n* = 113). **d** Comparison of NT50 of COVID-19 plasma for SARS-CoV-2 and SARS-CoV neutralization. SARS-CoV-2 or SARS-CoV pseudoviruses were pre-incubated with COVID-19 plasma from all severity groups (*n* = 104), 293-ACE2 cells were infected and RFP expression was determined at day 3 using flow cytometry. **e** Graph of SARS-CoV-2 NT50 values from hospitalized subjects plotted against SARS-CoV (*n* = 46). Two-tailed Mann–Whitney U test was used to determine the statistical significances in figures (**b**), (**c**) and (**d**) and two-tailed Spearman's was used for figure (**e**). Horizontal bars in (**b**), (**c**) and (**d**) indicate mean values.

SARS-CoV-2 and SARS-CoV pseudoviruses. Remarkably, most of the plasma samples also neutralized SARS-CoV pseudovirus, although less efficiently than SARS-CoV-2 pseudovirus (Fig. 5d). Interestingly, NT50 levels of plasma samples for SARS-CoV-2 and SARS-CoV significantly correlated in hospitalized subjects (Fig. 5e), whereas there were no correlations in outpatients and plasma donors or when all groups were combined (Supplementary Fig. 5a). In addition, when we compared the severity groups for their SARS-CoV NT50 values, we did not observe any significant difference between the groups (Supplementary Fig. 5b).

**Correlations of SARS-CoV-2 neutralization, antibody levels, and COVID-19 subject characteristics.** To better understand the associations between patient characteristics and the humoral immune response in COVID-19, we next determined correlations between antibody AUC levels, NT50 values and demographics of the study subjects. First, we assessed the correlation between NT50 values with S-RBD or Nucleocapsid antibody titers or their subclasses. All Igs including S-RBD IgG ($r_s = 0.81$), Nucleocapsid IgG ($r_s = 0.689$), S-RBD IgA ($r_s = 0.60$) and S-RBD IgM ($r_s = 0.47$) showed significant correlation with NT50 values of each subject (Fig. 6a). Among Ig subclasses specific to S-RBD; IgG1 ($r_s = 0.80$), IgG3 ($r_s = 0.69$) and IgG2 ($r_s = 0.67$) and, to a lesser degree, IgG4 ($r_s = 0.51$) also correlated with NT50 values (Fig. 6b). Total S-RBD IgG also correlated in a similar fashion

with other IgG isotypes, with IgG1 showing the highest positive correlation ($r_s = 0.96$) (Supplementary Fig. 4a).

Next, we correlated the antibody AUC levels and NT50 values of the subjects with their age. Subjects had significantly higher S-RBD IgG ($r_s = 0.53$), nucleocapsid IgG ($r_s = 0.43$), S-RBD IgA ($r_s = 0.41$) and NT50 ($r_s = 0.579$) values, as their age increased (Fig. 6c).

We also explored the relationship of the number of days between PCR test result and blood draw with antibody levels or NT50 values, excluding the subjects that had 15 days or fewer between those dates to ensure that antibody levels had already reached their peak. Of note, there was no correlation between the number of days and the IgG to S-RBD or the NT50 (Supplementary Fig. 4b), suggesting a potential persistence in antibody titers at least for the 3-month duration in this cohort.

## Discussion

The COVID-19 pandemic is continuing to spread globally unabated, including within the United States. There is an urgent need to better understand the immune response to the virus so that effective immune-based treatments and vaccines can be developed[21,22]. Neutralization of the virus by antibodies (NAbs) is one of the goals to achieve protection against SARS-CoV-2[23]. Despite the rapid development of many serological tests, important questions about the quality and quantity of seroprevalence in individuals still remain unclear[24,25]. Here, we

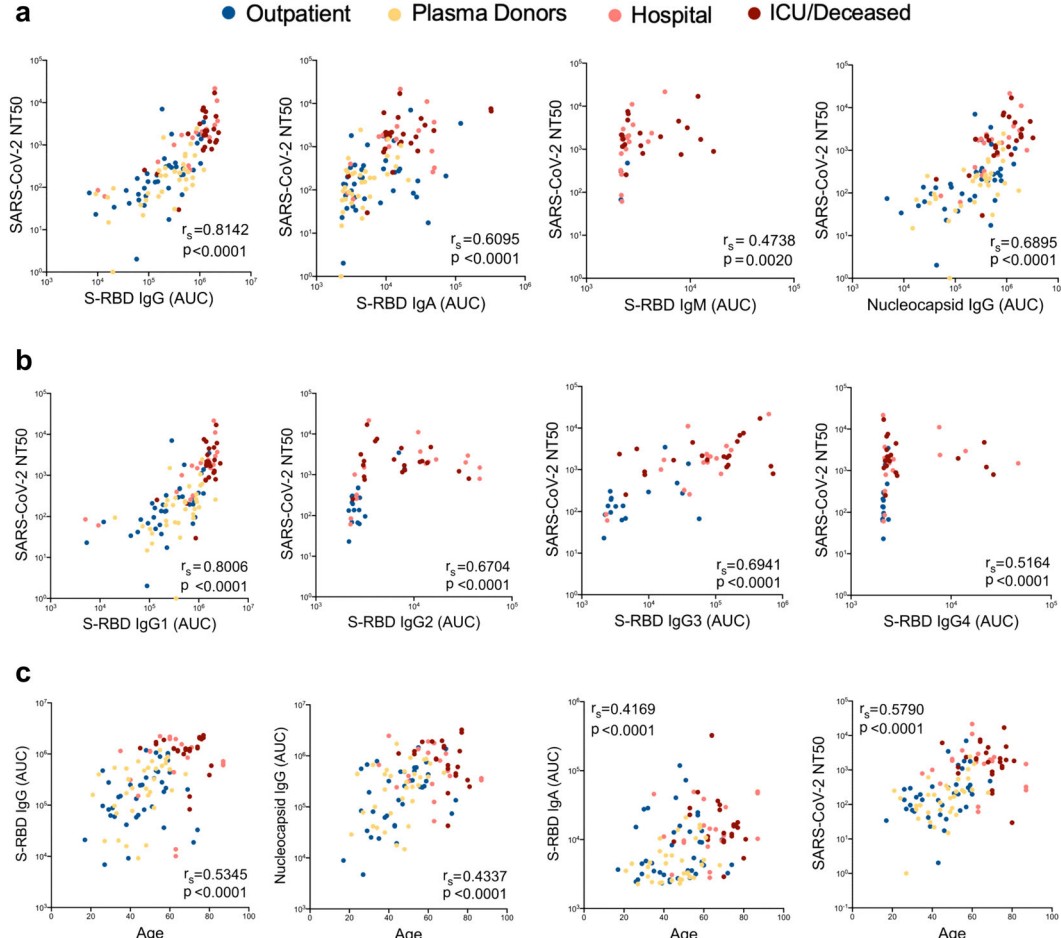

**Fig. 6 Correlations of antibody, neutralization levels and COVID-19 subject characteristics. a** Neutralization (NT50) of COVID-19 plasma correlated with S-RBD IgG ($n = 113$), S-RBD IgA ($n = 113$), S-RBD IgM ($n = 40$), and nucleocapsid IgG ($n = 113$). **b** Correlation of NT50 with S-RBD-specific IgG subclasses; IgG1 ($n = 113$), IgG2 ($n = 74$), IgG3 ($n = 74$), and IgG4 ($n = 74$). **c** Correlation of S-RBD IgG ($n = 115$), nucleocapsid IgG ($n = 115$), S-RBD IgA ($n = 115$), and NT50 ($n = 113$) with age. Two-tailed Spearman's was used to determine statistical significances.

developed highly sensitive and specific humoral assays that measure both the magnitude and neutralization capacity of antibody responses in COVID-19 patients. Every SARS-CoV-2 infected subject we tested ($n = 115$) had detectable antibodies and all subjects except one exhibited neutralization; both of these qualities were completely absent in non-infected controls. However, there was a profound difference in antibody and neutralization titers among subjects, ranging in more than 1000-fold differences. Furthermore, we found that almost all COVID-19 patients also had NAbs for SARS-CoV, suggesting a high degree of cross-reactivity between the spike proteins of these two viruses.

One of our key findings was clustering of antibody responses based on the severity of the disease; as hospitalized patients showed much higher antibody levels and neutralization capacity than outpatient subjects or convalescent plasma donors. This finding is consistent with recent reports suggesting that patients with more severe disease contain relatively higher levels of antibodies for SARS-CoV-2 infection[2,26–30]. Interestingly, most of the convalescent plasma donors had much lower levels of NAbs (by at least an order of magnitude) than hospitalized patients, who would be the suitable recipients for such plasma transfer therapy. This finding raises the question of whether convalescent plasma transfers may actually provide benefit to severe COVID-19 patients by providing NAbs. It may perhaps be more beneficial to identify donors with much higher NAb titers for the plasma donation. As such, our findings point to the importance of having access to assays that have a large dynamic range to detect antibody responses in COVID-19 patients or seropositive individuals. This neutralization assay also revealed differences in commercial antibodies to SARS-CoV-2 in their capacity to block virus entry, and as such can be used for rapid identification or generation of synthetic NAbs. In addition to measuring neutralization titers, the pseudoviruses can be used to probe cells that have the potential to be infected with SARS-CoV-2, given lentiviruses can infect most cell types and do not require cell division to integrate into the genome. This infection assay may also be used to screen small molecules that may impact virus cell entry.

Along with SARS-CoV-2, we also developed a pseudotyped lentivirus with SARS-CoV spike protein, which was equally efficient at infecting ACE2 overexpressing cells. This finding is consistent with results that SARS-CoV-2 spike protein in complex binding with human ACE2 (hACE2) is similar overall to that observed for SARS-CoV[31]. There was however slightly better neutralization of SARS-CoV-2 cell entry than SARS-CoV with soluble ACE2, which could be due to key residue substitutions in SARS-CoV-2, creating a slightly stronger interaction and thus higher affinity for receptor binding than SARS-CoV spike protein[31]. Accordingly, we also tested the ability of COVID-19 patient plasma for SARS-CoV neutralizing capacity and found significant SARS-CoV specific neutralization in COVID-19 patients. However, the neutralization titers for SARS-CoV were significantly lower and there was no correlation with the neutralization activity against SARS-CoV-2 when all severity groups were combined. Indeed, some donors even had relatively higher SARS-CoV neutralization (Fig. 5e). Further, there was a significant correlation between SARS-CoV-2 and SARS-CoV NT50 levels when hospitalized and ICU/deceased subjects were analyzed separately. Given that the two viruses share ~75% identical amino acid sequences in their spike proteins and there are conserved epitopes between them[32], it is conceivable that some of the SARS-CoV-2 NAbs have cross-neutralizing activity[33,34]. It is noteworthy that a recent study showed a NAb developed for SARS-CoV was highly effective at neutralizing SARS-CoV-2[35]. It is also tempting to speculate that presence of SARS-CoV specific NAbs could be more potent perhaps by targeting highly conserved regions of the spike protein, making it more difficult for the virus to select for escape mutants.

Other antibodies and neutralization assays have been developed during the submission of our manuscript[2,3,9,36–40]. Although direct comparison is difficult due to differences in the assay methodologies and different sample sets, we believe our assays embody distinctive features that further enhance this critically important immune response to SARS-CoV-2. The use of the flow cytometry bead-based fluorescent system that detects spike or Nucleocapsid protein-bound antibodies provides a high-throughput assay with a very high dynamic range and sensitivity, as it could detect antibodies from some subjects at up to a million-fold dilution of the plasma. This assay is also scalable and can be easily adaptable to other viral antigens. Using a flow cytometry platform is also important in that the assay can be further developed in a single panel to identify all antibody isotypes simultaneously and to complement flow cytometric immune phenotyping of COVID-19 patients. The high sensitivity and specificity of our assay has allowed us to correlate the spike protein RBD-specific antibody levels with neutralization titers, which showed very high concordance, and thus can be utilized as a proxy for neutralization in a clinical setting. Furthermore, the bead-based immunoassay can also be further developed to screen for antibodies reacting to other SARS-CoV-2 antigens simultaneously and can be useful in identifying antibodies that cross-react between different species of coronavirus proteins. Determining other isotypes such as IgA and IgG subclasses may also help in future mechanistic studies. It is clear that the dominant antibody response in almost all donors was IgG1, but some also show high IgA and IgG2-4, at varying levels. For example, given the importance of IgA antibodies in providing immunity on mucosal surfaces within the respiratory system, SARS-CoV-2 RBD-spike protein-specific IgA levels may also play an important role in the upper and lower respiratory system, or perhaps also in the gut, of COVID-19 patients[41,42].

There are also several potential practical implications for our findings. First, the patient population with the greatest risk factors for severe outcomes from infection such as age and co-morbid conditions had the highest antibody titers as well as neutralization of the virus. This is also the case for those patients who had a lethal disease. It is therefore possible that surviving COVID-19 may require non-antibody-dependent factors or that producing too much antibody may even have deleterious effects[43], such as potential antibody-dependent enhancement phenomenon by triggering Fc receptors on macrophages[44]. In this regard, it is interesting to note that a Bruton tyrosine kinase inhibitor that targets Fc-receptor signaling in macrophages is being tested in a randomized clinical trial[45]. Thus, understanding the mechanism of survival from COVID-19 and immune response dynamics will be critical in the better prediction of outcomes as well as in assaying for a protective response to potential vaccines.

In conclusion, the assays developed herein can have utility in uncovering dynamic changes in the antibody levels in SARS-CoV-2 infected subjects over time, in responses to vaccines and as potential clinical determinants for plasma or antibody therapies for COVID-19 patients.

## Methods

**Participants.** We enrolled 69 female and 46 male COVID-19 subjects with an average age of 54 and 51, respectively. The subjects ($n = 115$) were recruited at SUNY Downstate Medical Center, New York, NY, Cedars Sinai, Los Angeles, CA, or the University of Connecticut, School of Medicine, Farmington CT following testing and/or admission for COVID-19 infection. Written informed consent was obtained from all participants in this study and was approved by the following IRBs: (1) IRB# SUNY:269846. The patients were recruited at SUNY Downstate, NY and processed and biobanked at Amerimmune, Fairfax VA; (2) IRB#

STUDY00000640. Convalescent plasma was collected at Cedars Sinai Medical Center according to FDA protocol (https://www.fda.gov/vaccines-blood-biologics/investigational-new-drug-ind-or-device-exemption-ide-process-cber/recommendations-investigational-covid-19-convalescent-plasma#Collection%20of%20COVID-19). The source of the convalescent plasma was volunteer blood donors who were recovered from COVID-19. Donors met routine blood donor eligibility requirements established by the FDA and had a prior SARS-CoV-2 infection documented by a laboratory test for the virus during illness, or antibodies to the virus after recovery of suspected disease. All donors were at least 28 days from either resolution of COVID-19 symptoms or diagnostic clearance, whichever was longer; (3) IRB# 20-186-1. UConn Healthcare workers who tested positive for the virus by PCR were recruited and samples banked for future testing. (4) IRB#: 17-JGM-13-JGM or 16-JGM-06-JGM. De-identified control subjects ($n = 56$) with previously frozen (more than a year ago) samples obtained from healthy controls or determined to be SARS-CoV-2 PCR negative (IRB SUNY:269846). All antibody assays were performed at the Jackson Laboratory for Genomic Medicine, Farmington, CT. Subject characteristics are shown in Table 1. All plasma samples were aliquoted and stored at −80 °C. Prior to experiments, aliquots of plasma samples were heat-inactivated at 56 °C for 30 min.

**Over-expression of SARS-CoV-2 spike protein and cell culture.** Human codon-optimized SARS-CoV-2 spike-protein sequence was synthesized by Molecular-Cloud (MC_0101081). 5′-ACGACGGAATTCATGTTCGTCTTCCTGGTCCTG-3′ and 5′-ACGACGGAATTCTTAACAGCAGGAGCCACAGC-3′ primers were used to amplify the SARS-CoV-2 spike protein sequence without ERRS that is the last 19 amino acids[33]. Full length and truncated SARS-CoV-2 spike-protein sequences were then cloned into pLP/VSVG plasmid from Thermo Fisher under CMV promoter after removing the VSVG sequence via EcoRI-EcoRI restriction digestion. HEK-293T cells (ATCC; mycoplasma-free low passage stock) were transfected with the expression plasmids using Lipofectamine 3000 (Invitrogen) according to the manufacturer's protocol. The cells were cultured in complete RPMI 1640 medium (RPMI 1640 supplemented with 10% FBS; Atlanta Biologicals, Lawrenceville, GA), 8% GlutaMAX (Life Technologies), 8% sodium pyruvate, 8% MEM vitamins, 8% MEM nonessential amino acid, and 1% penicillin/streptomycin (all from Corning Cellgro) for 72 h[46]. The cells were then collected using %0.05 Trypsin-0.53 mM EDTA (Corning Cellgro) and stained with Biotinylated Human ACE2/ACEH Protein, Fc,Avitag (Acro Biosystems) then stained with APC anti-human IgG Fc Antibody clone HP6017 (Biolegend). Samples were acquired on a BD FAC-Symphony A5 analyzer and data were analyzed using FlowJo (Tree Star).

**Pseudotyped lentivirus production and titer measurement.** Lentivector plasmids containing RFP or GFP reporter genes were co-transfected with either SARS-CoV-2 spike protein or SARS-CoV spike protein (Human SARS coronavirus (SARS-CoV) spike glycoprotein Gene ORF cDNA clone expression plasmid (Codon Optimized) from SinoBiological) plasmids into HEK-293T cells using Lipofectamine TM 3000 (Invitrogen) according to the manufacturer's protocol. Viral supernatants were collected 24–48 h post-transfection, filtered through a 0.45 μm syringe filter (Millipore) to remove cellular debris, and concentrated with Lenti-X (Invitrogen) according to the manufacturers' protocol. Lentivirus supernatant stocks were aliquoted and stored at −80 °C. To measure viral titers, virus preps were serially diluted on ACE2 over-expressing 293 cells. Seventy-two hours after infection, GFP or RFP positive cells were counted using flow cytometry and the number of cells transduced with virus supernatant was calculated as infectious units/per mL.

**Generating human ACE2 over-expressing cells.** Wildtype ACE2 sequence was obtained from Ensembl Gene Browser (Transcript ID: ENST00000252519.8) and codon-optimized with SnapGene by removing restriction enzyme recognition sites that are necessary for subsequent molecular cloning steps, preserving the amino acid sequence. The sequence of mKO2 (monomeric Kusabira-Orange-2[47]) obtained from Addgene (#54625)[48], and was added onto the C terminal of ACE2 proximal to the stop codon with a small linker peptide (ccggtcgccacc) encoding the amino acids 'PVAT'. The fusion constructs were synthesized via GenScript and cloned into a lentiviral vector lacking a fluorescent reporter. The full-length human ACE2 sequence without fusion fluorescent proteins was amplified from the ACE2-mKO2 fusion construct using 5′-ACGACGGCCGGCCGCCATGTCAAGCTCTTCCTGGC-3′ and 5′-ACGACGGAATTCTTAAAAGGAGGTCTGAACATCATCAG-3′ primers, generating a stop codon at the C-terminus, and then cloned into a lentiviral vector encoding GFP reporter separated from multiple cloning site via an internal ribosome entry site (IRES) sequence. To determine the virus titers, HEK-293T cells were transduced with full-length ACE2-IRES-GFP, ACE2-mKO2 fusion construct lentiviruses and analyzed via flow cytometry for their reporter gene expression 72 h after infection. Wild type and ACE2 over-expressing HEK-293T were also stained with SARS-CoV-2 S1 protein, Mouse IgG2a Fc Tag (Acro Biosystems) followed with APC Goat anti-mouse IgG2a Fc Antibody (Invitrogen). Samples were acquired on a BD FACSymphony A5 analyzer and data were analyzed using FlowJo (BD Biosciences).

**SARS-CoV-2 antibody detection using flow immunoassay.** To screen for antibody binding to SARS-CoV-2 proteins, The DevScreen SAv Bead kit (Essen BioScience, MI) was used. Biotinylated 2019-nCoV (COVID-19) spike protein RBD, His, Avitag, Biotinylated CoV-2 (COVID-19) Nucleocapsid protein, His, Avitag, Biotinylated CoV-2 (COVID-19) S1 protein, His,Avitag (Acro Biosystems, DE), Biotinylated SARS-CoV-2 (2019-nCoV) Spike S1 NTD-His & AVI recombinant protein and Biotinylated SARS-CoV-2 (2019-nCoV) Spike S2 ECD-His recombinant protein (Sino Biological Inc.) were coated to SAv Beads according to manufacturer's instructions. Confirmation of successful bead conjugation was determined by staining with anti-His Tag (Biolegend) and flow cytometry analysis. S-RBD, N, S1, S1-NTD, and S2-ECD conjugated beads were then used as capture beads in flow immunoassay where they were incubated with anti-S-RBD human IgG positive control (provided in GenScript SARS-CoV-2 Spike S1-RBD IgG & IgM ELISA Detection Kit as positive control), recombinant Human ACE2-Fc (Acro Biosystems) or plasma and serum samples for 1 h at room temperature. Plasma samples were assayed at a 1:100 starting dilution and three additional tenfold serial dilutions. Anti-S-RBD antibody and ACE2-Fc were both tested at a 5 μg/mL starting concentration and in additional fivefold serial dilutions. Detection reagent was prepared using Phycoerythrin-conjugated anti-human IgG Fc clone HP6017, anti-human IgM clone MHM-88 (Biolegend), anti-human IgA clone IS11-8E10, anti-human IgG1 clone IS11-12E4.23.20 (Miltenyi Biotec), anti-human IgG2 Fc clone HP6002, anti-human IgG3 Hinge clone HP6050, and anti-human IgG4 pFc clone HP6023 (Southern Biotech), added to the wells and incubated for another hour at room temperature. Plates were then washed twice with PBS and analyzed by flow cytometry using iQue Screener Plus (IntelliCyt, MI). Flow cytometry data were analyzed using FlowJo (BD biosciences). DevScreen SAv Beads were gated on FSC-H/SSC-H, and singlet beads gate was created using FSC-A/FSC-H. Gates for different DevScreen SAv Beads were determined based on their fluorescence signature on RL1-H/RL2-H plot (on iQue plus). PE fluorescence median, which is directly associated with each single plex beads was determined using BL2-H (on iQue plus). Geometric means of PE fluorescence in different titrations were used to generate the titration curve and 20 healthy control plasma were used to normalize the AUC. Statistical analyses were performed using GraphPad Prism 8.0 software (GraphPad Software).

**SARS-CoV-2 antibody detection using ELISA.** To evaluate antibodies binding to CoV-2 S-RBD protein, SARS-CoV-2 Spike S1-RBD IgG and IgM ELISA Detection Kit from GenScript was used according to the manufacturer's instructions. Absorbance was measured at 450 nm using an Epoch 2 microplate spectrophotometer (BioTek).

**Pseudotyped virus neutralization assay.** Threefold serially diluted monoclonal antibodies including anti-SARS-CoV-2 Neutralizing human IgG1 antibody from Acro Biosystems, NAb#1 (Fig. 4c, d), GenScript clone ID 6D11F2, NAb#2 (Fig. 4d) and GenScript clone ID 10G6H5, NAb#3 (Fig. 4d), Invitrogen clone ID MA5-35939 Nab#4 (Fig. 4d), SARS-CoV-2 Spike S1-RBD IgG&IgM ELISA Detection Kit non-neutralizing antibody (Fig. 4c), recombinant human ACE2-Fc (Acro Biosystems, sACE2#1 and GenScript, sACE2#2 (Fig. 4d) or plasma from COVID-19 convalescent individuals and healthy donors have incubated with RFP-encoding SARS-CoV-2 or GFP-encoding SARS-CoV pseudotyped virus with 0.2 multiplicity of infection for 1 h at 37 °C. The mixture was subsequently incubated with 293-ACE2 cells for 72 h after which cells were collected, washed with FACS buffer (1 × PBS + 2% FBS) and analyzed by flow cytometry using BD FACSymphony A5 analyzer. Total cells were gated using FSC-A/SSC-A, GFP and RFP cell populations were gated using GFP-A/PE-A. Cells that do not express GFP or RFP were used to define the boundaries between positive and negative cell populations. To report the percent infection in ACE2 positive cells, total cells were sub-gated on GFP+ and RFP+ cell population was determined using PE-A.

Percent infection obtained for samples derived from cells infected with SARS-CoV-2 or SARS-CoV pseudotyped virus in the absence of plasma, ACE2-Fc or monoclonal antibodies. The half-maximal inhibitory concentration for plasma (NT50), ACE2-Fc or monoclonal antibodies (IC50) was determined using 4-parameter nonlinear regression (GraphPad Prism 8.0).

**Statistics and reproducibility.** All statistical analyses were performed using GraphPad Prism V8 software. Continuous variable datasets were analyzed by the Mann–Whitney $U$ test for non-parametric datasets when comparing clinical groups, and exact P values are reported. $P < 0.05$ was considered significant. Spearman p analysis was used to determine the relationship existing between two sets of non-parametric data, where a value of 0 indicated no relationship, values between 0 and ±0.3 indicated a weak relationship, values between ±0.3 and ±0.7 indicated a moderate relationship, values between ±0.7 and ±1.0 indicated a strong relationship, and a value of ±1.0 indicated a perfect relationship between sets of data[49]. We report the result of all considered pairwise tests, and thus make no formal adjustment for multiple comparisons. Numbers of repeats for each experiment were described in the associated figure legends.

**Reporting summary**. Further information on research design is available in the Nature Research Reporting Summary linked to this article.

## Data availability

The source data for the Figures along with the Supplementary Figures presented in this paper are available as Supplementary Data 1.

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

## Acknowledgements

The research in this study was supported by the National Institute of Health (NIH) grants R01AI121920, U54 NS105539, and U19 AI142733 to D.U. We thank Tina Vaziri for critical reading. We would like to acknowledge Drs. Kevin Dieckhaus, Mark Metersky, Eric Mortensen, Robert Fuller, and David Banach for their advice and consultation.

## Author contributions

M.D., L.K. and D.U. conceived and designed the experiments. M.D., L.K., L.P., M.Y. and R.H. carried out the experiments. B.T.L. designed the clinical research study on UConn Healthcare workers and M.K. recruited participants and executed clinical protocols. R.G. and O.A. designed the clinical research study at SUNY State Medical Center, M.P., Z.B., M.B., P.C. and K.L., recruited, processed, coordinated, and executed the clinical protocols. M.A., E.K., N.M. and C.H., designed, recruited, and stored convalescent plasma samples at Cedars Sinai hospital. M.D., L.K, C.L.G. and D.U. analyzed the data, performed statistics, drew illustrations and prepared the figures. M.D. and D.U. wrote the manuscript.

## Competing interests

The authors declare no competing interests.
