## [Peer Review File · Communications Biology]

Reviewers' comments:

Reviewer #1 (Remarks to the Author):

Dogan et al report a new, efficient method for serological testing of SARS-CoV-2 infected individuals based on the use of fluorescent bead assays. Such assays are broadly used and well renowned for their versatility, and dynamic range as well multiplex capacity. This is an important advancement and much needed to mitigate the current pandemic. The authors also report interesting relationships between disease severity and serologic responses.

Some specific questions that could be addressed.

1. What is the exact specificity and sensitivity estimates for the anti-RBD IgG/IgA and IgM assays?
2. One hypothesis proposed to explain severe COVID-19 disease is Antibody-mediated enhancement. Can the authors comment on this given their neutralization data?
3. Have the authors been able to compare patients of different age-groups? in particular it would be interesting to assess IgG-responses in children who seem to experience almost exclusively mild infections.

Reviewer #2 (Remarks to the Author):

In the present manuscript, Dogan et al. describe two methods developed in their lab to measure SARS-CoV-2 antibodies and the neutralization potency of patient sera. They assessed anti-RBD and anti-nucleocapsid IgG (and its subclasses), IgA and IgM in patients with different severity of COVID-19 using a flow immunoassay. Neutralizing potency was measured using pseudoviruses synthesized from lentivectors with full length CoV-2 spike or SARS-CoV-1 spike. The manuscript is clear and provides valuable information on the differences in antibody titers and neutralizing potential between patients with mild or more severe disease. However, some of the findings described in this manuscript, e.g. the development of the neutralization assay or the positive correlation between clinical severity and antibody titers, were described in previous publications and unfortunately provide limited novel information.

Comments:

1. The flow immunoassay data looks promising. It is however difficult to extrapolate whether the method truly has a higher sensitivity and specificity compared to other gold standard methods (e.g. ELISA) as this comparison is not provided. A more comprehensive study of different methods is expected to confirm the superiority of the new method.
2. By coating beads with RBD or the nucleoprotein only, the authors might miss a lot of antibodies that would bind the spike protein outside of the RBD. For instance, previous studies have shown that antibodies binding the NTD could exhibit high neutralization potency against SARS-CoV-2. The authors could evaluate the titers of binding antibodies to other fragments of the spike protein such as the full extracellular domain, S2 and NTD.
3. Could the immobilization of the biotinylated protein onto the beads hide epitopes for antibody binding?

4. The authors point out the negative correlation between the number of days between the PCR test and the blood draw and values for IgG/A to RBD, IgG to N and NT50, indicating a potential decline in titers over time. Such a statement might be misleading in the present case, as all patients with the longest time between the PCR test and the blood draw are recovered patients (“plasma patients”) who might have had lower antibody titers at any time point (if the positive correlation between clinical severity and antibody titers is confirmed). Information about the illness severity of these patients would also be important to add.

5. Figure 2 shows the same data in different panels (e.g. A vs C, D, E, F). Showing panels A and B, and the difference between males and females (F) would be sufficient.

6. The authors assessed the neutralization potency of sera from hospitalized patients against SARS-CoV. Were healthy controls and other patients with lower SARS-CoV-2 titers also included? In addition, it would be interesting to compare this finding with anti-SARS-CoV antibodies in the serum as well.

7. Neutralization assays using lentivirus backbones and CoV-2 spike plasmids have already been widely described in previous publications (e.g.: doi: 10.3390/v1205051, doi: 10.1084/jem.20201181). How the neutralization assay described here would be more sensitive and of higher throughput than previous methods is not clear and should be addressed.

8. The neutralizing monoclonal antibodies used in the neutralization assay are human (NAb #3) or mouse (NAb #1 and 2). The neutralizing potency from 2 different species should not be compared or at least should be addressed in the manuscript. Also, which NAb was used for Figure 4C should be mentioned.

9. The following additional information should be provided in the methods section:

- Which pCMV vector was used?
- What was the SARS-CoV-1 plasmid?
- How much virus was added to the sera or mAb?
- Under “generating human ACE2 over-expressing cells”: why were WT and ACE2 293T cells stained with mouse IgG2a Fc tag?

10. I thought the use of “SARS” in the manuscript can be at first confusing. I would suggest mentioning “SARS-CoV” (or “SARS-CoV-1”) to avoid confusion.

xxx

...Further communication with reviewer 2 about detracting literature:

Here are some references I would suggest:

Regarding the development of a neutralization assay:

- <https://pubmed.ncbi.nlm.nih.gov/32692348/>
- <https://pubmed.ncbi.nlm.nih.gov/32384820/>
- <https://pubmed.ncbi.nlm.nih.gov/32207377/>

Regarding clinical severity and antibody titers:

I believe there is still ongoing research on this, therefore the information provided by Prof. Unutmaz and colleagues about the differences in antibody titers and neutralizing potential between patients with mild or more severe disease is still relevant. It seems that IgG appears sooner in severe cases, and that neutralizing antibody titers are higher in severe cases.

- <https://pubmed.ncbi.nlm.nih.gov/32634129/>
- <https://pubmed.ncbi.nlm.nih.gov/32555424/>
- <https://pubmed.ncbi.nlm.nih.gov/32350462/>
- <https://pubmed.ncbi.nlm.nih.gov/32221519/>

xxx

Reviewer #3 (Remarks to the Author):

This is a thorough and timely study of quantity, quality, and duration of human polyclonal antibody responses in the individuals with SARS-CoV2 infection. Dogan et al. used newly developed flow cytometric beads binding assays for screening of COVID-19 patient or convalescent plasma samples from 87 individuals. They determined antigen specificity and subclasses of polyclonal antibodies that were elicited in response to SARS-CoV-2 infection and compared the antibody titers between hospitalized COVID-19 individuals, outpatients, and convalescent plasma donors. The authors also compared antibody neutralization titers between the same cohorts of individuals using newly developed SARS-CoV-2 Spike protein pseudotyped lentivirus neutralization assays. Hospitalized individuals revealed higher IgG antibody responses and higher neutralization titers compared to outpatient or convalescent plasma donors. The authors have also suggested practical implications for their beads-based immunoassay for SARS-CoV-2 antibody detection. Overall, the results are of high interest to the field and have implications for understanding the host humoral response to SARS CoV-2 and, potentially, for diagnostic. There are some comments below, which if addressed, would make the paper clearer and more accurate.

Comments

1. For “Antibodies... thought to play an important role in the protective immune response to CoV-2 infection” in the Introduction – there are several key studies showing antibody-mediated protection in animal models of SARS-CoV-2 infection. The list of few: PMID: 32425270; 32553273; 32668443; 32404477; 32698192.
2. The paper title highlights novel binding and neutralization assays for SARS-CoV-2, and in the Introduction, it stated that “...currently available tests lack dynamic range and sensitivity...”. There have now been multiple reports for the development and validation of neutralization assays for SARS-CoV-2, including the wild-type virus, BSL3-luciferase reporter live virus, replication recombinant VSV expressing COV-2 S, as well as several pseudovirus antibody-based neutralization assays. The list of few: PMID: 32540903; 32555388; 32561270; 32735849; 32668443; 32651581; 32540900 etc. The authors should mention those studies and highlight any differences and similarities with their neutralization assays.
3. The statistics section is missing from the Materials and Methods.
4. For the Fig.2C-F, and Fig.5B-C, the p values are somewhat misleading to readers as the authors made a series of pair-wise statistical tests without correcting for multiple comparisons. Please, revise the analysis or clarify the rationale for reporting the uncorrected p values. Please, define the error bars and horizontal bars (mean or median?) in the legend.
5. Please, define the error bars and number of replicates in the legend for Fig.3F.

6. The authors have concluded that “We also observed measurable differences in the neutralizing activity of three different Nabs and 2 different soluble Ace2 proteins from two different sources (Fig. 4d), showing the utility of this assay...” – the neutralization curves in Fig.4 C-D are lacking the error bars that would not allow to compare the curves or to conclude about the assay accuracy and reproducibility. Is this only one measurement for each compound or data are the average of several measurements (technical replicates)? Please, add the error bars to the plots in Fig. 4C, D and Fig.S3B (if replicates are available), clarify representations in the legend, and revise the conclusion if applicable.

7. For the Spearman correlation analysis in Fig. 6 and Fig. S4, please, indicate in statistics section of the Materials and Methods the r-value ranges that are considered by the authors as a strong, moderate, and weak or no correlation.

8. In the Discussion the authors hypothesized that “It is possible that surviving COVID-19 may require non-antibody depended factors or that producing too much antibody may even have deleterious effects”. There have now been several reports showing association of higher neutralizing antibody titers in individuals with more severe or lethal COVID-19 disease. However, it is unclear if the higher titer of neutralizing antibodies is a primary reason of more severe disease, or a consequence from higher viral load and immune system stimulation in the diseased individuals. Thus, the other studies have clearly demonstrated that passive transfer of potent neutralizing antibodies alone or vaccination with spike protein induces a robust humoral-response-mediated immunity against SARS-CoV-2. The prophylactic and therapeutic efficacy of monoclonal antibody treatments has also been shown. The list of few: PMID: 32434945; 32668443; 32645327; 32553273; 32404477. The authors should discuss these studies.

9. The key for Figure S4 is missing (the color-coding of the human samples based on disease severity).

Pavlo Gilchuk, Ph.D.
Sr. Staff Scientist
Vanderbilt Vaccine Center

The Editor
Nature Communications Biology

October 12, 2020

Dear Editor,

We are submitting the revised version of our manuscript entitled "**Novel SARS-CoV-2 specific antibody and neutralization assays reveal wide range of humoral immune response during COVID-19**" by Dogan et al. for publication in *Nature Communications Biology*.

We greatly appreciate and grateful to the constructive critiques of all three Reviewers. To address all the critiques, we performed new experiments, increased our subject number to 115 and also added more negative controls, added several new figures (**Figures 1D, 1E, 5E, Supplementary Figures 1B and 5**) and revised others (**Figures 2, 3F, 4C, 4D, 5B, 5C, 5D, 6, Supplementary Figures 1A, 1B, 1C, 2A, 3, 4, and 5**) and extensively revised the manuscript accordingly. All new significant additions or edits to the text are **yellow marked** to make it easier for reviewers to determine the changes. Together, we believe these confirmed our key points in the original submission and greatly enhanced the impact of our findings.

Below is our point-by-point response to all the critiques:

Reviewer: 1

Dogan et al report a new, efficient method for serological testing of SARS-CoV-2 infected individuals based on the use of fluorescent bead assays. Such assays are broadly used and well renowned for the ir versatility, and dynamic range as well multiplex capacity. This is an important advancement and much needed to mitigate the current pandemic. The authors also report interesting relationships between disease severity and serologic responses.

1. What is the exact specificity and sensitivity estimates for the anti-RBD IgG/IgA and IgM assays?

We thank the reviewer for the positive comments. The statistical sensitivity and specificity estimates of our bead-based antibody assays were 100% and 99.34% for S-RBD IgG; 100% and 90.9% for S-RBD IgM, 94.26% and 87.87% for S-RBD IgA and 99.13% and 94.93% for Nucleocapsid IgG, respectively. We have now added these estimates to revised manuscript.

2. One hypothesis proposed to explain severe COVID-19 disease is Antibody-mediated enhancement. Can the authors comment on this given their neutralization data?

It is now well established that during COVID-19 collateral damage caused by severe response of immune system is partly responsible for the pathology. In our study, severely ill subjects had higher neutralization titers, that is also significantly correlated with their antibody levels. However, higher neutralization levels and high severity of the disease appear to be conflicting features of these subjects given that high neutralization capacity should counteract the viral pathology. Immune pathology such as antibody-mediated enhancement, complement activation

and subsequent collateral damage to tissues due to over-activation of immune system might perhaps contribute to the severity of the disease groups. As such, our observations remain correlative.

3. Have the authors been able to compare patients of different age-groups? in particular it would be interesting to assess IgG-responses in children who seem to experience almost exclusively mild infections.

We have performed analysis of the age-groups, however this analysis is confounded by the fact that most in the severe patient group were older subjects, thus potentially skewing the findings of antibody levels. We have not yet analyzed sufficient numbers of children with SARS-Cov-2 to answer this important question, we hope to have that as a separate study in near future.

Reviewer: 2

In the present manuscript, Dogan et al. describe two methods developed in their lab to measure SARS-CoV-2 antibodies and the neutralization potency of patient sera. They assessed anti-RBD and anti-nucleocapsid IgG (and its subclasses), IgA and IgM in patients with different severity of COVID-19 using a flow immunoassay. Neutralizing potency was measured using pseudoviruses synthesized from lentivectors with full length CoV-2 spike or SARS-CoV-1 spike. The manuscript is clear and provides valuable information on the differences in antibody titers and neutralizing potential between patients with mild or more severe disease. However, some of the findings described in this manuscript, e.g. the development of the neutralization assay or the positive correlation between clinical severity and antibody titers, were described in previous publications and unfortunately provide limited novel information.

We thank the reviewer for the positive comments. The reviewer is correct about our neutralization assay being not the first of its kind in the field and some clinical correlations which were described in publications after our submission of the manuscript.

We feel that our study is unique in combining highly efficient pseudotyped virus assay (both for SARS-CoV-2 and SARS-CoV-1) and very sensitive unique antibody test with large dynamic range. It is worth noting that, we have developed some of the first pseudotyped lentiviruses more than 20 years ago and have very expertise in this field. We feel the efficiency, reproducibility and quality of these assays are important, as we have tried to extensively document in multiple figures, and our study can form a good example for comparison to similar studies, which were not as high efficiency similar to ours thus far.

In addition, we have now, with the new experiments during revision, been able to test the stability of both viruses after multiple freeze/thaw cycles and proved that they preserve the similar activity of infection after multiple cycles (**Fig. 3f**) Taken together, these experimentally proven features of our pseudotyped lentiviruses allow us to establish a reliable and robust neutralization assay.

Another important point we would like to emphasize about our neutralization assay is the high correlation of neutralization titers with antibody levels (**Fig. 6a**). We think this is due to the high

sensitivity of both assays and thus, the antibody assay could be used as a proxy to the neutralization assay in order to shorten the duration of humoral immune response analysis in similar studies.

Regarding the clinical correlations described previous studies, we would like to highlight the high sensitivity and wide dynamic range of our assays. As far as we are aware, the drastic amplitude of the dynamic range among the individuals who were infected by SARS-CoV-2 and have the similar disease severity was not previously reported in other studies in a fashion we have reported in our cohort. Indeed, we have been able to detect antibody responses in 100% of the PCR+ confirmed subjects and neutralization in 99% of these, whereas none of the controls showed cross reactive neutralization.

1. The flow immunoassay data looks promising. It is however difficult to extrapolate whether the method truly has a higher sensitivity and specificity compared to other gold standard methods (e.g. ELISA) as this comparison is not provided. A more comprehensive study of different methods is expected to confirm the superiority of the new method.

We thank the reviewer for this important critique. We have now compared the sensitivity and specificity of the bead-based assay with a commercial ELISA-based antibody assay by screening some of the plasma samples with both assays. Antibody levels from the two antibody assays showed a high correlation ($r_s = 0.86$), confirming our assay's precision, and the bead-based antibody assay showed a wider dynamic range compared to the ELISA-based assay. (**Fig. 1e**). We also estimated the sensitivity and specificity of ELISA-based antibody assay as 89.36% and 100%, respectively, whereas bead-based assay's sensitivity and specificity estimates were calculated as 100% and 100%, also confirming our overall estimates in all subjects which were 100% and 99.34% for S-RBD IgG, respectively. We also confirmed much greater dynamic range of our bead-based immunoassay compared to the ELISA as shown in the new data.

2. By coating beads with RBD or the nucleoprotein only, the authors might miss a lot of antibodies that would bind the spike protein outside of the RBD. For instance, previous studies have shown that antibodies binding the NTD could exhibit high neutralization potency against SARS-CoV-2. The authors could evaluate the titers of binding antibodies to other fragments of the spike protein such as the full extracellular domain, S2 and NTD.

We again thank the reviewer for this suggestion which lead us to do further evaluations on our antibody assay. In addition to S-RBD and Nucleocapsid protein, we have now also attached different viral components such as S1 subunit of Spike protein, S1 subunit N Terminal Domain (NTD) and S2 Extracellular Domain (ECD) onto the magnetic beads and tested IgG levels specific to those viral proteins to compare the antibody levels they detect. Interestingly, S-RBD captured significantly more antibodies compared to S1, and had higher mean value compared to the other subunit parts we tested (**Fig. 1d**).

3. Could the immobilization of the biotinylated protein onto the beads hide epitopes for antibody binding?

Immobilization of the viral components could possibly hide some epitopes for antibody binding however, we feel that the strong correlations of antibody levels with each other and with NT50 values confirm that bead-based assay is able to detect most potential functionally relevant antibodies.

4. The authors point out the negative correlation between the number of days between the PCR test and the blood draw and values for IgG/A to RBD, IgG to N and NT50, indicating a potential decline in titers over time. Such a statement might be misleading in the present case, as all patients with the longest time between the PCR test and the blood draw are recovered patients ("plasma patients") who might have had lower antibody titers at any time point (if the positive correlation between clinical severity and antibody titers is confirmed). Information about the illness severity of these patients would also be important to add.

We thank the reviewer for this comment, which prompted us to correlate the number of the days between positive PCR date and blood drawn and the antibody levels or NT50 values in different severity groups separately and now also included more PCR+ subjects in the analysis. When we separated these groups, indeed, there was no significant correlation (**Supplementary Fig. 4b**). We are hoping to obtain second time points in some of these subjects at 6 month and after, which should be more revealing about the persistence of the antibodies. However, with our current results we conclude that the antibody levels do not appear to change more than physiological drop post disease for at least 3-month period. Other newer studies with more sensitive tests also appear to confirm this finding.

5. Figure 2 shows the same data in different panels (e.g. A vs C, D, E, F). Showing panels A and B, and the difference between males and females (F) would be sufficient.

The reason why we included **Figure C, D and E in Figure 2** is to emphasize the exact statistical differences between the severity groups as well as to clearly show the dramatic differences even among the patients who have the same disease severity. We felt it was not as clear to make our key points otherwise, thus would like to retain these separate figures for the clarity.

6. The authors assessed the neutralization potency of sera from hospitalized patients against SARS-CoV. Were healthy controls and other patients with lower SARS-CoV-2 titers also included? In addition, it would be interesting to compare this finding with anti-SARS-CoV antibodies in the serum as well.

We thank the reviewer for this critical point. We now performed experiments with all of the subjects in the manuscript for neutralization of SARS-CoV and compared this to SARS-CoV-2 in the new revised figures (**Fig. 5d, e**). We found that, most of the COVID19 plasma samples also neutralized SARS-CoV pseudovirus, although less efficiently than SARS-CoV-2 pseudovirus. Interestingly, NT50 levels of plasma samples for SARS-CoV-2 and SARS-CoV significantly

correlated in hospitalized subjects (**Fig. 5e**), whereas there were no correlations in outpatients and plasma donors or when all groups were combined (**Supplementary Fig. 5a**).

We also agree with the reviewer on the suggestion stating it would be interesting to compare this finding with anti-SARS-CoV antibodies in the serum but we have not been able to set up a system in which we capture the anti-SARS-CoV antibodies in the scope of this study. However, we think the NT50 values for SARS-CoV of COVID19 patients would provide us somewhat similar findings.

7. Neutralization assays using lentivirus backbones and CoV-2 spike plasmids have already been widely described in previous publications (e.g.: doi: 10.3390/v1205051, doi: 10.1084/jem.20201181). How the neutralization assay described here would be more sensitive and of higher throughput than previous methods is not clear and should be addressed.

We could not find the first paper, perhaps the DOI was incorrect. The second paper was published on July 21st after our submission thus was missed in our initial manuscript version. The NT50 values of pseudoviruses in the published manuscript was measured via Luciferase activity, which is different than our fluorescence-based flow assay, thus it is difficult to compare directly. However, their results seem somewhat comparable to our assay in terms of sensitivity, but it would be impossible to make direct comment without testing the same subject samples.

8. The neutralizing monoclonal antibodies used in the neutralization assay are human (NAb #3) or mouse (NAb #1 and 2). The neutralizing potency from 2 different species should not be compared or at least should be addressed in the manuscript. Also, which NAb was used for Figure 4C should be mentioned.

We now changed this in the manuscript according to reviewer's suggestions. In **Figure 4d**, NAb #1 and #4 were human antibodies whereas NAb #2 and #3 were mouse. Nab#1 used in **Figure 4c**.

*9. The following additional information should be provided in the methods section:
- Which pCMV vector was used?*

Full length and truncated SARS-CoV-2 spike protein sequences were cloned into pLP/VSVG plasmid from ThermoFisher under CMV promoter after removing the VSVG sequence.

- What was the SARS-CoV-1 plasmid?

SARS-CoV Spike protein plasmid was Human SARS coronavirus (SARS-CoV) Spike glycoprotein Gene ORF cDNA clone expression plasmid (Codon Optimized) from SinoBiological.

- How much virus was added to the sera or mAb?

Viruses were used at limiting doses of 0.2 multiplicity of infection (MOI) for neutralization assays. We had previously tested different MOIs extensively and found this to be the most optimal/sensitive for detecting neutralization.

- Under “generating human ACE2 over-expressing cells”: why were WT and ACE2 293T cells stained with mouse IgG2a Fc tag?

Fc tag was attached to S1 subunit of SARS-CoV-2 Spike protein. SARS-CoV-2 S1 protein, Mouse IgG2a Fc Tag (Acro Biosystems) was used for staining of ACE2 expression.

All of this missing information is now added to the Methods and reagents section of the manuscript. We again thank the reviewer for reminding these important points.

10. I thought the use of “SARS” in the manuscript can be at first confusing. I would suggest mentioning “SARS-CoV” (or “SARS-CoV-1”) to avoid confusion.

We apologize in retrospect for the confusion. We have now changed all the names “CoV-2” to “SARS-CoV-2” and “SARS” to “SARS-CoV”

...Further communication with reviewer 2 about detracting literature: Here are some references I would suggest. Regarding the development of a neutralization assay:

- <https://pubmed.ncbi.nlm.nih.gov/32692348/>
- <https://pubmed.ncbi.nlm.nih.gov/32384820/>
- <https://pubmed.ncbi.nlm.nih.gov/32207377/>

Regarding clinical severity and antibody titers, I believe there is still ongoing research on this, therefore the information provided by Prof. Unutmaz and colleagues about the differences in antibody titers and neutralizing potential between patients with mild or more severe disease is still relevant. It seems that IgG appears sooner in severe cases, and that neutralizing antibody titers are higher in severe cases.

- <https://pubmed.ncbi.nlm.nih.gov/32634129/>
- <https://pubmed.ncbi.nlm.nih.gov/32555424/>
- <https://pubmed.ncbi.nlm.nih.gov/32350462/>
- <https://pubmed.ncbi.nlm.nih.gov/32221519/>

We thank for these suggestions on literature, which we now included all these studies and mentioned them in the discussion section of our revised manuscript.

Reviewer: 3

This is a thorough and timely study of quantity, quality, and duration of human polyclonal antibody responses in the individuals with SARS-CoV2 infection. Dogan et al. used newly

developed flow cytometric beads binding assays for screening of COVID-19 patient or convalescent plasma samples from 87 individuals. They determined antigen specificity and subclasses of polyclonal antibodies that were elicited in response to SARS-CoV-2 infection and compared the antibody titers between hospitalized COVID-19 individuals, outpatients, and convalescent plasma donors. The authors also compared antibody neutralization titers between the same cohorts of individuals using newly developed SARS-CoV-2 Spike protein pseudotyped lentivirus neutralization assays. Hospitalized individuals revealed higher IgG antibody responses and higher neutralization titers compared to outpatient or convalescent plasma donors. The authors have also suggested practical implications for their beads-based immunoassay for SARS-CoV-2 antibody detection. Overall, the results are of high interest to the field and have implications for understanding the host humoral response to SARS CoV-2 and, potentially, for diagnostic. There are some comments below, which if addressed, would make the paper clearer and more accurate.

Comments

1. For “Antibodies... thought to play an important role in the protective immune response to CoV-2 infection” in the Introduction – there are several key studies showing antibody-mediated protection in animal models of SARS-CoV-2 infection. The list of few: PMID: 32425270; 32553273; 32668443; 32404477; 32698192.

We thank the reviewer for the kind positive comments. We have cited the mentioned studies in our revised manuscript.

2. The paper title highlights novel binding and neutralization assays for SARS-CoV-2, and in the Introduction, it stated that “...currently available tests lack dynamic range and sensitivity...”. There have now been multiple reports for the development and validation of neutralization assays for SARS-CoV-2, including the wild-type virus, BSL3-luciferase reporter live virus, replication recombinant VSV expressing COV-2 S, as well as several pseudovirus antibody-based neutralization assays. The list of few: PMID: 32540903; 32555388; 32561270; 32735849; 32668443; 32651581; 32540900 etc. The authors should mention those studies and highlight any differences and similarities with their neutralization assays.

We have added these to the discussion, with the caveat that direct comparison is difficult due to differences in the assay methodologies and of course different sample sets.

3. The statistics section is missing from the Materials and Methods.

We thank the reviewer for reminding this missing section. We have now added the Statistics section in Materials and Methods.

4. For the Fig.2C-F, and Fig.5B-C, the p values are somewhat misleading to readers as the authors made a series of pair-wise statistical tests without correcting for multiple comparisons. Please, revise the analysis or clarify the rationale for reporting the uncorrected p values. Please, define the error bars and horizontal bars (mean or median?) in the legend.

Pair-wise statistical tests in mentioned figures (two-tailed Mann Whitney U tests) were performed to compare 2 groups at a time. Given we are only comparing two groups comparatively, we believe this would not require us to correct p values separately. However, we

have now defined the error bars and horizontal bars in the figure legends. Horizontal bars indicate mean values.

5. Please, define the error bars and number of replicates in the legend for Fig.3F.

We now defined the error bars and number of replicates in the legend for **Figure 3f**. Titration experiments were replicated twice except for the “1 freeze/thaw cycle” for which titrations were replicated 4 times. Error bars represent 1 standard deviation of mean values.

6. *The authors have concluded that “We also observed measurable differences in the neutralizing activity of three different Nabs and 2 different soluble Ace2 proteins from two different sources (Fig. 4d), showing the utility of this assay...” – the neutralization curves in Fig.4 C-D are lacking the error bars that would not allow to compare the curves or to conclude about the assay accuracy and reproducibility. Is this only one measurement for each compound or data are the average of several measurements (technical replicates)? Please, add the error bars to the plots in Fig. 4C, D and Fig.S3B (if replicates are available), clarify representations in the legend, and revise the conclusion if applicable.*

We have now revised the **Figure 4c** and **4d**, which represent 3 replicates of the experiments. Error bars indicate one standard deviation of mean values. We have removed **Figure S3B** since we believe **Figure 3SA** is sufficient to make the same point.

7. *For the Spearman correlation analysis in Fig. 6 and Fig. S4, please, indicate in statistics section of the Materials and Methods the r-value ranges that are considered by the authors as a strong, moderate, and weak or no correlation.*

We have indicated in the Statistics section of Materials and Methods that a value of 0 indicated no relationship, values between 0 and +/-0.3 indicated a weak relationship, values between +/-0.3 and +/-0.7 indicated a moderate relationship, values between +/-0.7 and +/-1.0 indicated a strong relationship, and a value of +/-1.0 indicated a perfect relationship between sets of data, also cited article about mentioned ranges.

8. *In the Discussion the authors hypothesized that “It is possible that surviving COVID-19 may require non-antibody depended factors or that producing too much antibody may even have deleterious effects”. There have now been several reports showing association of higher neutralizing antibody titers in individuals with more severe or lethal COVID-19 disease. However, it is unclear if the higher titer of neutralizing antibodies is a primary reason of more severe disease, or a consequence from higher viral load and immune system stimulation in the diseased individuals. Thus, the other studies have clearly demonstrated that passive transfer of potent neutralizing antibodies alone or vaccination with spike protein induces a robust humoral-response-mediated immunity against SARS-CoV-2. The prophylactic and therapeutic efficacy of monoclonal antibody treatments has also been shown. The list of few: PMID: 32434945; 32668443; 32645327; 32553273; 32404477. The authors should discuss these studies.*

We thank the reviewer for this important point and agree that it is not clear whether higher antibody titers are the primary reason of severe disease or a consequence. We speculated this as a possibility in the severe patient group, which may also have other genetic or immunological predispositions, such as uncontrolled macrophage or complement activation via antibody Fc-

triggering. It is also important to note that, as shown in our study, plasma donors had lower antibody levels as these are mixed individuals with mild/moderate/severe disease and more time has passed since they have been tested positive. Thus, plasma transfer may never reach the extremely high levels of antibodies present in severe patients.

9. The key for Figure S4 is missing (the color-coding of the human samples based on disease severity).

We apologize for this omission and have placed the key for **Figure S4** in the revised manuscript.

REVIEWERS' COMMENTS:

Reviewer #2 (Remarks to the Author):

The answers and corrections to the revised version of "Novel SARS-Cov-2 specific antibody and neutralization assays reveal wide range of humoral immune response during COVID-19", by Dogan et al., significantly improved the manuscript. I believe that Dogan et al. have addressed all my questions and concerns in their revised version. The manuscript is clear, the data well presented and convincing. I believe this manuscript is of high interest. I have no other concerns or questions.

Reviewer #3 (Remarks to the Author):

The manuscript has been largely improved and the authors addressed my comments in a satisfactory manner.

Minor comments:

1. Since some data set comparisons indicated "ns" (not significant), please specify a significance level (e.g. $p < 0.05$ was considered significant) in the Statistics section of Materials and Methods.
2. For the authors response "Given we are only comparing two groups comparatively, we believe this would not require us to correct p values separately"- please, add a statement to the Methods section "We report the result of all considered pairwise tests, and thus make no formal adjustment for multiple comparisons".

Pavlo Gilchuk, Ph.D.

Sr. Staff Scientist

Vanderbilt Vaccine Center